# Waldenström’s Macroglobulinemia and Cryoglobulinemic Glomerulonephritis: An Unusual Case of Monoclonal Gammopathy of Renal Significance

**DOI:** 10.3390/medsci11040077

**Published:** 2023-12-05

**Authors:** José C. De La Flor, Jesús de María Sulca, Pablo Rodríguez, Daniel Villa, Edna Sandoval, Rocío Zamora, Maribel Monroy-Condori, Roxana Lipa, Henry Perez, Michael Cieza

**Affiliations:** 1Department of Nephrology, Hospital Central Defense Gomez Ulla, 28047 Madrid, Spain; 2Department of Nephrology, Hospital Cayetano Heredia, Lima 15002, Peru; jesus.sulca@upch.pe (J.d.M.S.); henry.perez.s@upch.pe (H.P.); michael.cieza@upch.pe (M.C.); 3Department of Nephrology, Guadalajara Center Dialysis, AVERICUM, 19003 Guadalajara, Spain; pablo.rodriguez@avericum.com; 4Department of Nephrology, Clínica Universidad de Navarra, 31008 Navarra, Spain; devillah@unav.es; 5Department of Hematology, Hospital Central Defense Gomez Ulla, 28047 Madrid, Spain; esanba5@mde.es; 6Department of Nephrology, Hospital Universitario General Villalba, 28400 Madrid, Spain; rocio.zamora@hgvillalba.es; 7Department of Nephrology, Hospital General Universitario Nuestra Señora del Prado, 45600 Talavera de la Reina, Spain; mmonroy@sescam.jccm.es; 8Department of Anatomical Pathology, Instituto Nacional de Salud del Niño-San Borja, Lima 15037, Peru; rlipa@insnsb.gob.pe

**Keywords:** Waldenström’s macroglobulinemia, cryoglobulinemia glomerulonephritis, monoclonal gammopathy of renal significance

## Abstract

Cryoglobulins are immunoglobulins that precipitate at temperatures below 37 °C and dissolve upon reheating. They can induce small-vessel vasculitis with renal involvement. Cryoglobulinemic glomerulonephritis is a rare manifestation that occurs in patients with monoclonal gammopathy, specifically Waldenström’s macroglobulinemia. We present the case of a 52-year-old patient with a history of cutaneous vasculitis and hypothyroidism, who presented with generalized edema, moderate anemia, hypercholesterolemia, nephrotic range proteinuria of 12.69 g/day, microhematuria, arterial hypertension, and hypocomplementemia via the classical pathway, without acute kidney injury and with negative serological studies and positive cryoglobulins in the second determination. Serum and urine protein electrophoresis and immunofixation studies showed a monoclonal band of IgM and kappa light chain. Renal biopsy was consistent with cryoglobulinemic glomerulonephritis. In the context of dysproteinemia and cryoglobulinemic glomerulonephritis, bone-marrow aspiration and biopsy were performed, leading to the diagnosis of Waldenström’s macroglobulinemia. Monoclonal gammopathies have been described in association with type I cryoglobulinemias. This described association is uncommon, which is why we present this case, along with a review of the literature.

## 1. Introduction

The cryoglobulins (CGs) are immunoglobulins (Igs) that precipitate at temperatures below 37 °C and dissolve upon reheating [1]. They can induce cryoglobulinemic vasculitis (CryoVas), which affects small and medium-sized vessels, and is mediated by immune complexes (IC) [2]. CGs are classified into three subtypes: type I consists of isolated monoclonal Igs (IgM) and is often associated with plasma cell dyscrasias; type II comprises monoclonal IgM with positive rheumatoid factor (RF) and polyclonal IgG; and type III includes polyclonal IgG and IgM with positive RF [3]. Types II and III are termed mixed cryoglobulinemias (MC). The most common clinical manifestations affect the skin, joints, peripheral nervous system, and kidneys, with the latter being affected in 30% of cases. Cryoglobulinemic glomerulonephritis (CGN) is the most common renal involvement of CryoVas, especially in the context of MC and hepatitis C virus (HCV) infection, occurring in approximately 90% of cases, while the remaining 10% are associated with autoimmune diseases and mainly B-cell lymphoproliferative neoplasms. Patients typically present with nephritic syndrome and arterial hypertension (AHT). CGN is a histopathological entity that can exhibit various proliferative or sclerosing patterns on light microscopy (LM), with membranoproliferative glomerulonephritis (MPGN) being the most common histological pattern, occurring in 70–90% of cases [2].

Waldenström’s Macroglobulinemia (WM) is a rare form of non-Hodgkin’s lymphoma (NHL), accounting for only 1–2% of hematologic malignancies. It is characterized by the infiltration of the bone marrow by clonal lymphoplasmacytic cells, leading to the production of circulating monoclonal immunoglobulin M (IgM) [4]. WM follows a diverse clinical course, with symptoms and complications often linked to the tumor’s size or quantity, as well as the physicochemical or immunologic properties of the monoclonal IgM [5,6]. Less than 8% of patients with WM present with renal involvement. However, various glomerular and tubular lesions have been described in patients with WM [7]. Immunoglobulin light chain (AL) amyloidosis and CGN are the two predominant glomerular pathologies in WM. The term monoclonal gammopathy of renal significance (MGRS) was introduced by the International Kidney and Monoclonal Gammopathy Research Group [8]. Monoclonal IgM gammopathies (MG IgM) fall within the framework of monoclonal B-cell lymphocytosis. Most of them are MGRS, while others may progress to WM, and less frequently to non-Hodgkin B-cell lymphomas. Herein, we present an unusual case of a patient with CGN associated with WM.

## 2. Case Presentation

A 52-year-old woman with a history of hypothyroidism secondary to radioactive iodine treatment for hyperthyroidism, and unclassified leukocytoclastic cutaneous vasculitis treated with oral corticosteroids for 6 months, presented to the emergency department with general malaise, uncontrolled AHT, generalized edema, and foamy urine over a 5-month period. On physical examination, she had a blood pressure of 150/80 mmHg, respiratory rate of 24 breaths per minute, edema in the abdomen, face, and lower limbs with 4+ pitting edema up to the thighs. No purpuric lesions, arthralgias, splenomegaly, Raynaud’s phenomenon, or palpable nodules were noted. Laboratory tests revealed mild normocytic normochromic anemia with thrombocytosis, normal haptoglobin, reticulocytes 0.9%, peripheral blood smear without schistocytes, negative Coombs test and rheumatoid factor. Urea/creatinine dissociation at 80 with normal levels of urea (U) and creatinine (Cr). Additionally, she had severe hypoalbuminemia, hypercholesterolemia, and nephrotic range proteinuria. Urinalysis showed sterile leukocyturia and hematuria of 25 red blood cells per field. Serological tests were negative for Hepatitis B, HCV, and HIV. Hypocomplementemia C3 and C4 with normal autoimmunity study and serum cryoglobulins were negative in the first determination and positive in the second. Serum protein electrophoresis (SPEP) and serum immunofixation electrophoresis (SIFE) were similar to the urinary protein electrophoretic/immunofixation (UPEP/UIFE) analysis, revealing a monoclonal IgM-kappa (κ) band. Serum free light chain kappa (FLCκ) and lambda (FLCλ) were 330.16 mg/dL and 15.81 mg/L, respectively, with a κ/λ FLC ratio of 20.88. Renal ultrasound showed increased size and echogenicity in both kidneys. The rest of the laboratory results are shown in Table 1.

Given this clinical picture, a renal biopsy (RB) was performed. Light microscopy (LM) revealed 16 glomeruli, none of which were sclerosed, with accentuation of lobular architecture, double-contoured capillary loops, and hyaline thrombi staining positively with periodic acid-Schiff (PAS) in the capillary lumen (Figure 1A,B). Congo red staining was negative. Direct immunofluorescence (IF) was mainly positive for IgM (2+) (Figure 2A), C3 (+) (Figure 2B) and kappa (2+) (Figure 2E), localized in capillary loops and mesangium and more intensely in the pseudothrombus (Figure 2A–E). Electron microscopy (EM) showed deposits of unorganized immune complexes, predominantly subendothelial, and amorphous cylinders in the capillary lumen within the pseudothrombus without substructures (Figure 3A–D). Findings were consistent with a histological pattern of MPGN with cellular crescent in one glomerulus, with atypical immune deposits of IgM-kappa subendothelially in the mesangium and in the pseudothrombi, compatible with a type I CGN. Bone marrow aspiration (BMA) revealed preserved cellularity and no blasts or cells foreign to the bone marrow. Bone marrow biopsy (BMB) showed cellularity of 45% with apparent integrity of the three main series. There was an interstitial increase in plasma cells and lymphocytes. Immunohistochemistry concluded medullary infiltration by low-grade lymphoplasmacytic lymphoma with plasma cell differentiation. Based on the clinical, analytical, and histological findings from the skin biopsy, BMA, BMB, and RB, in conjunction with the rheumatology, hematology, and nephrology services, a diagnosis of systemic CryoVas with type I CGN secondary to WM was established, meeting criteria for treatment due to renal involvement. The patient initiated chemotherapy with six cycles of Bortezomib/Dexamethasone (BordD). Bortezomib was administered at a dose of 1.3 mg/m^2^ given as an SC bolus on days 1, 8, and 15. She received Rituximab in the second cycle of chemotherapy at a dose of 1 gr on days 1 and 15, after which she experienced tumor lysis syndrome. Following management of the acute episode, she continued with her chemotherapy with BordD. After six cycles of treatment with BordD, a complete renal response was achieved, and partial hematologic response was achieved with a reduction of more than 50% in kappa light chain levels.

## 3. Discussion

We present an unusual case of WM with MPGN-type renal involvement and a monoclonal IgM deposit, consistent with a CGN. Most patients with WM exhibit symptoms and signs related to tumor infiltration (hepatosplenomegaly), monoclonal protein in serum (cryoglobulinemia) and in tissues (amyloidosis), and/or antibody production (neuropathy with hemolytic anemia). Our patient did not present hepatomegaly, which is seen in 20% of WM patients, nor did she have lymphadenopathy, observed in 15% of cases [7]. Our patient presented with nephrotic/nephritic syndrome without acute kidney injury (AKI). It has been described that AHT in the context of nephritic syndrome is the most prevalent manifestation (>50%) in patients with CGN [9]. Higgins et al. described the renal manifestations of WM, defining CGN as the presence of endocapillary proliferation or MPGN with abundant intracapillary monocytic infiltrate and at least one of the following findings: intraluminal glomerular pseudothrombi, focal deposits with characteristic cryoglobulin substructures in EM, and/or positive serum cryoglobulins [10]. Our patient presented positive serum cryoglobulins in a second determination, associated with an RB showing a histological pattern of MPGN and pseudothrombi. In a study by Higgins et al. [10], 1363 WM patients participated, of which 57 had RB and BMB examination. Renal histology showed CGN as the second most frequent renal lesion (28%), after AL amyloidosis (33%). The latter was part of the differential diagnosis in our case; however, it was discarded as the RB showed a negative Congo red stain. A histological pattern of MPGN with pseudothrombi in RB and hypocomplementemia via the classical pathway could be a manifestation of lupus nephritis (LN) as well; however, the patient did not meet the EULAR/ACR classification criteria for systemic lupus erythematosus [11].

The most frequent finding on IF on CGN in patients with WM is the presence of dominant monoclonal IgM staining, with specificity for kappa or lambda light chains, involving the mesangium, the walls of glomerular capillaries, or intraluminal pseudothrombi [7]. The IF in our case described 2+ positivity in loops, mesangium, and capillary pseudothrombi for IgM and kappa light chains, like the findings described in the literature. It is important to mention that in cases of monoclonal B-cell lymphocytosis, other hematological diseases may also exhibit monoclonal IgM. Therefore, monoclonal IgM gammopathy is not an exclusive indicator of WM [7]. Most of these cases are monoclonal gammopathy of undetermined significance (MGUS), though others may evolve into WM, and less frequently into non-Hodgkin B-cell lymphomas. Previous studies have shown significant heterogeneity in renal manifestations associated with monoclonal IgM gammopathy, and there are very few available data on the evolution of these patients and their renal prognosis. LN could lead to the development glomerular thrombi, which occur in 41 to 49% of cases. Therefore, it is important to differentiate hyaline thrombi caused by deposits (pseudothrombi) from actual thrombi. IF staining is very useful for this identification. If the deposit, such as a thrombus, shows negative fibrin staining and positive Ig staining, the latter component depending on the type of cryoglobulin, it is a cryoglobulinemic pseudothrombus; the opposite result supports a real thrombus [12]. In our case, there was positive staining for fibrinogen in 3/12 glomeruli, which excluded the possibility of the patient having actual glomerular thrombi.

Uppal et al. [7] described subendothelial deposits with or without substructural organization in EM, which may have fibrillar and microtubular features. It is important to mention that the massive glomerular intracapillary deposits of monoclonal IgM that occlude the capillary lumen have been recognized for decades as a “hallmark” of renal involvement secondary to WM [13]. The EM in our patient revealed deposits of unorganized immune complexes located on the subendothelium and amorphous cylinders in the capillary lumen, without describing substructures, neither the fibrillar nor microtubular types. Yi-Pu Chen et al. [12] noted that deposits in mesangium, subendothelium, and capillary lumen are frequently present. These may exhibit a short fibrillar or microtubular substructure. Meanwhile, deposits containing cryoglobulins may show an annular or highly organized microtubular substructure. Nonetheless, it is worth noting that not all EM images of RB from patients may reveal such substructures. Therefore, the diagnosis of CGN should not be ruled out solely based on the absence of these lesions [12].

On the other hand, Khwaja et al. [14] recently reported on the clonal and clinical characteristics of a cohort of 134 patients who were monitored for 3 years with IgM type I cryoglobulins. Among these, 76% were diagnosed with WM, 5% had other types of NHL, and 19% had IgM MGUS. In cases where cryoglobulins and coexisting cold agglutinin disease (CAD) were present, the molecular characteristics were consistent with a CAD clone, with wild-type MYD88 observed in 80% of cases. At the time of cryoglobulin detection, half of the patients exhibited active manifestations, with vasomotor symptoms being the most common (22%) and hyperviscosity being the least common (9%). Treatment for cryoglobulin-related symptoms alone was required in 12% of cases. Notably, hyperviscosity emerged as the most significant predictor of treatment-related cryoglobulinemia or death (HR: 73.01; 95% CI: 15.62–341.36; *p* < 0.0001). As a result, the authors conclude that type I IgM cryoglobulinemia is more prevalent than previously recognized in cases of IgM gammopathy and should be actively pursued in clinical evaluation.

There is no universally standardized approach for managing WM. Treatment initiation should be based on the severity of symptoms or complications. It is recommended when symptoms are associated with hyperviscosity or venous thrombosis, or if the disease has expanded with organ involvement. Plasma exchange is useful in cases of acute kidney injury, hyperviscosity syndrome, clinical manifestation of cryoglobulinemia, and antibody-mediated damage. Our patient was treated with furosemide, hydrochlorothiazide, and spironolactone, which allowed for the control of edema and blood pressure. After the diagnosis of WM with CGN, chemotherapy with Dexamethasone/Bortezomib was started. No plasma exchange was performed due to clinical improvement and the absence of acute kidney injury and symptoms related to hyperviscosity. Upon the occurrence of tumor lysis syndrome after Rituximab administration in the second chemotherapy cycle, the previously established treatment was continued. The use of proteasome inhibitors, such as Bortezomib, in the management of WM showed improved clinical and laboratory responses in more than 75% of patients in a study by Chen et al. [15]. Additionally, new therapies targeting the pathological clone in the management of WM with renal involvement are known, which could help decrease this complication in the future. According to the consensus panel of the 11th International Workshop on WM, current practice recommendations suggest considering first-line treatment options for symptomatic, treatment-naïve patients. These options include rituximab-based chemoimmunotherapy (CIT), dexamethasone, cyclophosphamide, rituximab (DRC), bendamustine, or rituximab (Benda-R). Covalent inhibitors of Bruton’s tyrosine kinase (cBTKi) present a well-tolerated alternative, particularly for primary treatment in MW patients, especially those who are ineligible for CIT [16].

## 4. Conclusions

In conclusion, we describe the clinical, analytical, and histological manifestations of an unusual case of CGN secondary to WM, a rare disease whose physiopathology of renal involvement is not yet fully understood. Histopathological identification within the context of a hematological lymphoproliferative process plays a crucial role in determining the optimal therapeutic approach. It is imperative for patients with MGRS to undergo a comprehensive evaluation by a multidisciplinary team consisting of nephrologists, hematologists, and nephropathologists to elucidate the causative role of the M protein in the pathogenesis of renal disease.

## Figures and Tables

**Figure 1 medsci-11-00077-f001:**
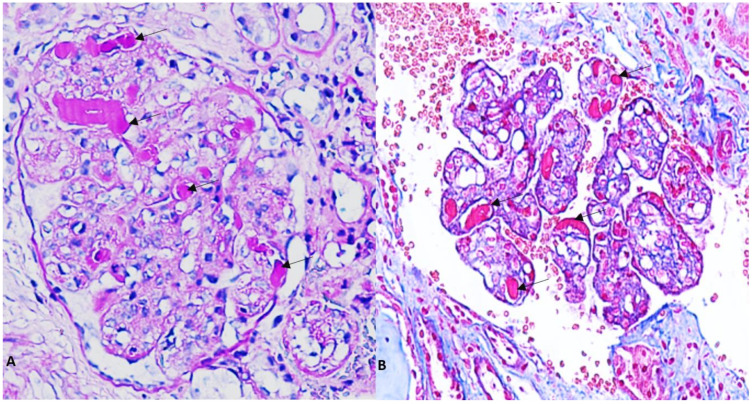
(**A**). Light microscopy showing a glomerulus with intracapillary pseudotrombi (indicatedby arrows) positive for PAS staining (PAS Stain ×400). (**B**). Pseudotrombi with negative Trichrome staining (Masson’s Trichrome Stain ×400).

**Figure 2 medsci-11-00077-f002:**
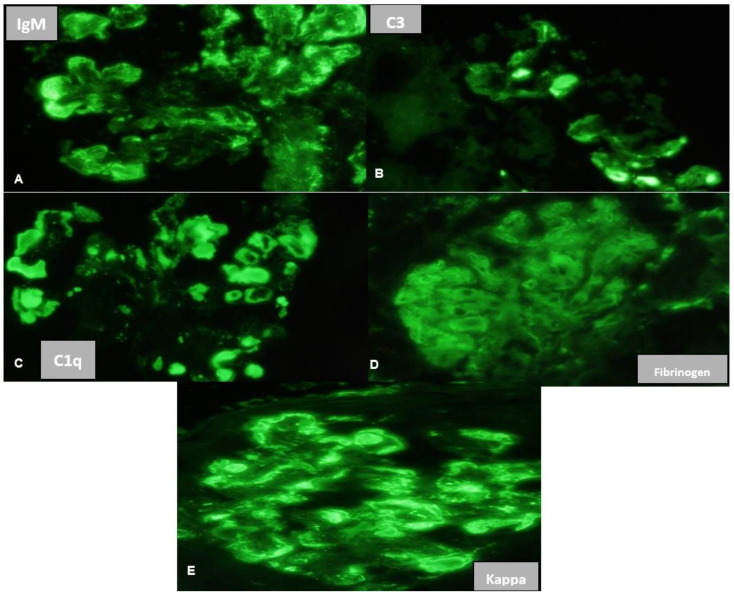
Immunofluorescence Staining. (**A**). IgM: Positive 2+ in loops, mesangium, and thrombi in 12/12 glomeruli. (**B**). C3: Positive 1+ in loops and in thrombi in 7/12 glomeruli. (**C**). C1q: Positive 1+ in loops and in thrombi in 7/12 glomeruli. (**D**). Fibrinogen: positive +/− in loops in 3/12 glomeruli. (**E**). Kappa: positive 2+ in loops and thrombi in 12/12 glomeruli.

**Figure 3 medsci-11-00077-f003:**
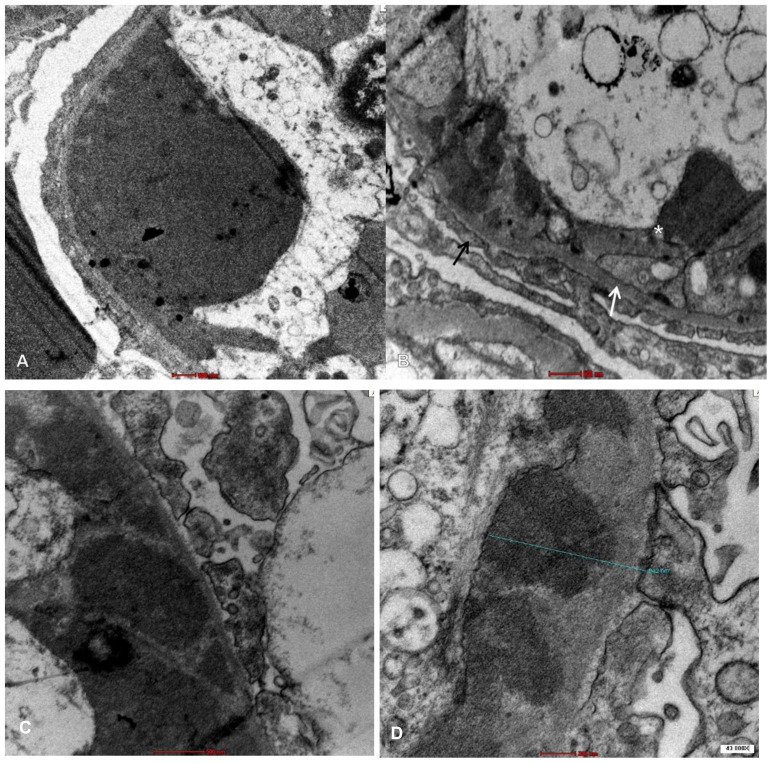
Electron Microscopy. (**A**). Presence of pseudothrombus in the capillary lumen of a glomerulus. (**B**). Subendothelial deposits are evident in electron microscopy. The black arrow points to podocyte processes, the white arrow indicates the glomerular basement membrane, and the white asterisk marks a subendothelial deposit. (**C**). Subendothelial deposits in electron microscopy. (**D**). Subendothelial deposit measuring 942 nm at 43,000× magnification.

**Table 1 medsci-11-00077-t001:** Analytical Parameters Upon Admission.

		Normal Range—Units
White blood cells	8100	4–10 × 10^3^ μL
Hemoglobin	9.7	12–16 g/dL
Platelets	649 × 10^3^	10^3^/μL
Reticulocytes	0.9%	0.5–2%
Lactate dehydrogenase	126	135–214 IU/L
Coombs test	Negative	NA
Total bilirubin	0.2	0.1–1 mg/dL
Total proteins	4.1	6.4–8.7 g/dL
Serum albumin (Alb)	2	3–5.5 g/dL
AST	34	5–32 IU/L
ALT	7.2	5–33 IU/L
Urea	48	17–60 mg/dL
Creatinine	0.6	0.6–1.2 mg/dL
Na^+^	129	135–145 mmol/L
K^+^	4.8	3.5–5.5 mmol/L
Cl^−^	94	95–110 mmol/L
C-Reactive protein	Negative	0.1–0.5 mg/dL
Total cholesterol	324	mg/dL
HBsAg (hepatitis B surface antigen)	Negative	NA
Anti-HCV antibodies	Negative	NA
HIV	Negative	NA
C3	69	90–180 mg/dL
C4	2	10–40 mg/dL
Rheumatoid factor	9	<15 IU/mL
Beta 2 microglobulin	2	<0–20 mg/dL
ANA, anti-dsDNA, ANCA	Negative	NA
Serum cryoglobulin	Positive	NA
Anti-PLA2R antibody (ELISA)	Negative	NA
Urine red blood cells	5–10	/HPF
24 h urine total protein excretion	12690.4	<0.15 g/24-h
IgG	460	800–1600 mg/dL
IgA	120	70–400 mg/dL
IgM	775	90–180 mg/dL
UPEP/UIFE	IgM kappa	NA
SPEP M-protein concentration	6%	NA
SIFE	IgM kappa	NA
FLCκ	330.16	4.90–13.70 mg/L
FLCλ	15.81	7.60–19.50 mg/L
FLCκ/λ	20.88	0.27–1.67
Urine culture	Negative	NA

NA: Not applicable, AST: aspartate aminotransferase; ALT: alanine aminotransferase; Na^+^: sodium; Cl^−^: chlorine; K^+^: potassium; HIV: human immunodeficiency virus, HBsAg: hepatitis B surface antigen, HCV: hepatitis C virus; C3: complement factor C3; C4: complement factor C4; ANA: antinuclear antibodies; anti-DNAds: anti double-stranded DNA antibody; ANCA: antineutrophil cytoplasmic antibody; anti-PLA2R-Ab: anti-phospholipase A2 receptor antibody; Ig: immunoglobulins; UPEP/UIFE: urine protein electrophoresis/urine immunofixation electrophoresis; SPEP: serum protein electrophoresis; SIFE: serum immunofixation electrophoresis; FLCκ: free light chains kappa; FLCλ: free light chains lambda.

## Data Availability

No new data were created or analyzed in this study. The data used to support the findings of this study are available from the corresponding author on request (contact J.C.D.L.F., josedelaflor81@yahoo.com, jflomer@mde.es). I confirm that all the figures and tables are the original work of this manuscript’s authors. All figures and tables were produced by the authors of this manuscript; they were not adapted from other authors, and do not include an online link.

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
