# Peer review of "Waldenström’s Macroglobulinemia and Cryoglobulinemic Glomerulonephritis: An Unusual Case of Monoclonal Gammopathy of Renal Significance"

_medsci, 2023, doi:10.3390/medsci11040077_

Round 1

Reviewer 1 Report

Comments and Suggestions for Authors

A well presented case with excellent images.

Good introduction with useful context.

Well presented case, with all relevant clinical features cited.

Use of WM and MW- please be consistent.

Line 180-181: "Therefore IgM gammopathy is not a pathognomonic finding of WM' does not make sense as a sentence nor as a conclusion from the prior sentences. Needs rewording.

Sentence 208-9 does not make sense

Sentence 209-210: by cell lysis do the authors mean tumour lysis syndrome?

It would be good for the Discussion to be reworked and tightened up.

Author Response

COVER LETTER RESPONSE TO REVIEWERS AND EDITOR

Editor-in-Chief ¨Medicines¨ 

Dear Assigned Editor Ms. Nevena Uletilovic and Reviewers:

Thank you so much for your constructive suggestions and comments on our manuscript.

Please find enclosed our revised Manuscript ID medsci-2692803 entitled “Waldenström's Macroglobulinemia and Cryoglobulinemic Glomerulonephritis: An Unusual Case of Monoclonal Gammopathy of Renal Significance" which we would like to resubmit for publication in Medical Sciences Journal (MDPI),

The reviewer´s comments are all valuable and very helpful for improving our paper. We have studied these comments carefully and have made corrections accordingly, which we believe to have significantly improved the manuscript, and we hope that those changes will be met by your approval.

In the following pages, you will find our point-to-point responses to each of the reviewers’ comments:

Reviewer #1 report:

Comments to authors

A well-presented case with excellent images. Good introduction with useful context.

Well-presented case, with all relevant clinical features cited.

1.- Use of WM and MW- please be consistent.

Thank you very much for your comment. We will keep in mind not to make this mistake, we will modify MW to WM.

2.- Line 180-181: "Therefore IgM gammopathy is not a pathognomonic finding of WM' does not make sense as a sentence nor as a conclusion from the prior sentences. Needs rewording.

Thank you very much , it has been corrected.

3.- Sentence 208-9 does not make sense

Thank you very much , it has been corrected

4.- Sentence 209-210: by cell lysis do the authors mean tumour lysis syndrome?

 Thank you very much for your comment. If so, I will modify the phrase to tumor lysis syndrome.

5.- It would be good for the Discussion to be reworked and tightened up

Thank you very much for your comment, we have included a small paragraph about treatment  and revised the discussion.

In addition to this letter, we send a word file with a revised version of the manuscript with the changes made highlighted, and a clean version.

Sincerely yours,

Dr. De La Flor José

Corresponding author's address

Reviewer 2 Report

Comments and Suggestions for Authors

The authors describe a case of Waldenström’s macroglobulinemia with cryoglobulinemia, resulting in cryoglobulinemic glomerulonephritis. This case report may be a useful clinical reminder for general internists, nephrologists, hematologists, and pathologists. The case description is clear and concise; the discussion is adequate and well based on the observations, and the paper is well-written. I have only minor comments.

Minor comments:

The definition of Waldenström’s macroglobulinemia (Line 59-60) should be more exact.

1.       The statement that “There is no standard therapy for WM” should be softened. While it is true that there is no single, universal standard therapy, the International Workshop for Waldenström Macroglobulinemia (IWWM) regularly publishes updated therapeutic guidelines. This should be briefly mentioned, and a recent report should be referenced.

2.       Please classify the cryoglobulinemia in this specific patient into the appropriate type (1, 2, or 3).

3.       An original research article on the association between WW and cryoglobulinemia as well as the distribution of cryoglobulinemia types in WM has recently been published (Khwaja et al. Br J Haematol 2023; PMID: 37726004). I suggest this reference be included in the Reference List and briefly commented in relation to the authors’ observations.

Author Response

COVER LETTER RESPONSE TO REVIEWERS AND EDITOR

Editor-in-Chief ¨Medicines¨ 

Dear Assigned Editor Ms. Nevena Uletilovic and Reviewers:

Thank you so much for your constructive suggestions and comments on our manuscript.

Please find enclosed our revised Manuscript ID medsci-2692803 entitled “Waldenström's Macroglobulinemia and Cryoglobulinemic Glomerulonephritis: An Unusual Case of Monoclonal Gammopathy of Renal Significance" which we would like to resubmit for publication in Medical Sciences Journal (MDPI),

The reviewer´s comments are all valuable and very helpful for improving our paper. We have studied these comments carefully and have made corrections accordingly, which we believe to have significantly improved the manuscript, and we hope that those changes will be met by your approval.

In the following pages, you will find our point-to-point responses to each of the reviewers’ comments:

Reviewer #2 report:

Comments to authors

The authors describe a case of Waldenström’s macroglobulinemia with cryoglobulinemia, resulting in cryoglobulinemic glomerulonephritis. This case report may be a useful clinical reminder for general internists, nephrologists, hematologists, and pathologists. The case description is clear and concise; the discussion is adequate and well based on the observations, and the paper is well-written. I have only minor comments.

Minor comments:

1.- The definition of Waldenström’s macroglobulinemia (Line 59-60) should be more exact.

Thank you very much for your comment. I modify the paragraph of lines 59-61 ¨Waldenström's Macroglobulinemia (WM) is a rare B-cell lymphoma characterized by infiltration of lymphoplasmacytic cells in the bone marrow and other organs with the presence of monoclonal IgM in the serum¨ to read as follows:

¨Waldenström's Macroglobulinemia (WM) is a rare form of non-Hodgkin's lymphoma (NHL), accounting for only 1%-2% of hematologic malignancies. It is characterized by the infiltration of the bone marrow by clonal lymphoplasmacytic cells, leading to the production of circulating monoclonal immunoglobulin M (IgM) (4). WM follows a diverse clinical course, with symptoms and complications often linked to the tumor's size or quantity, as well as the physicochemical or immunologic properties of the monoclonal IgM (5, 6).¨ 

4.-Teras LR, DeSantis CE, Cerhan JR, Morton LM,Jemal A, Flowers CR. 2016 US lymphoidmalignancy statistics by World Health Organization subtypes. CA Cancer J Clin. 2016;66(6):443-459.

5.- Gayathri Ravi , Prashant Kapoor. Current approach to Waldenström Macroglobulinemia. Cancer Treat Res Commun. 2022:31:100527. doi: 10.1016/j.ctarc.2022.100527.

6.- Meletios A. Dimopoulos and Efstathios Kastritis. How I treat Waldenstrom macroglobulinemia. Blood. 2019 Dec 5;134(23):2022-2035. doi: 10.1182/blood.2019000725.

2.- The statement that “There is no standard therapy for WM” should be softened. While it is true that there is no single, universal standard therapy, the International Workshop for Waldenström Macroglobulinemia (IWWM) regularly publishes updated therapeutic guidelines. This should be briefly mentioned, and a recent report should be referenced.

Thank you very much, We have modified that comment for this one:

¨There is no universally standardized approach for managing WM. Treatment initiation should be based on the severity of symptoms or complications. It is recommended when symptoms are associated with hyperviscosity, venous thrombosis, or if the disease has extended with organ involvement¨

 We have added a paragraph about the recent recommendations of treatment of the International Workshop for Waldenström Macroglobulinemia (IWWM) 2021:

¨According to the consensus panel of the 11th International Workshop on WM, current practice recommendations suggest considering first-line treatment options for symptomatic, treatment-naïve patients. These options include rituximab-based chemoimmunotherapy (CIT), dexamethasone, cyclophosphamide, rituximab (DRC), or bendamustine, rituximab (Benda-R). Covalent inhibitors of Bruton Tyrosine Kinase (cBTKi) present a well-tolerated alternative, particularly for primary treatment in MW patients, especially those who are ineligible for CIT (16).¨

3.- Please classify the cryoglobulinemia in this specific patient into the appropriate type (1, 2, or 3).

Thank you very much for your comment. We classify according to clinical, analytical, and histological findings from skin biopsy, Bone marrow aspiration, Bone marrow biopsy and Renal Biopsy as a type I cryoglobulinemia.

4.-  An original research article on the association between WW and cryoglobulinemia as well as the distribution of cryoglobulinemia types in WM has recently been published (Khwaja et al. Br J Haematol 2023; PMID: 37726004). I suggest this reference be included in the Reference List and briefly commented in relation to the authors’ observations.

Thank you very much for recommending to add this original arithmetic, we have added this paragraph to the discussion:

¨On the other hand, Khwaja et al. (14) recently reported on the clonal and clinical characteristics of a cohort of 134 patients who were monitored for 3 years with IgM type I cryoglobulins. Among these, 76% were diagnosed with WM, 5% had other types of NHL, and 19% had IgM MGUS. In cases where cryoglobulins and coexisting cold agglutinin disease (CAD) were present, the molecular characteristics were consistent with a CAD clone, with wild-type MYD88 observed in 80% of cases. At the time of cryoglobulin detection, half of the patients exhibited active manifestations, with vasomotor symptoms being the most common (22%) and hyperviscosity being the least common (9%). Treatment for cryoglobulin-related symptoms alone was required in 12% of cases. Notably, hyperviscosity emerged as the most significant predictor of treatment-related cryoglobulinemia or death (HR: 73.01; 95% CI: 15.62-341.36; p<0.0001). As a result, the authors conclude that type I IgM cryoglobulinemia is more prevalent than previously recognized in cases of IgM gammopathy and should be actively pursued in clinical evaluation.¨

In addition to this letter, we send a word file with a revised version of the manuscript with the changes made highlighted, and a clean version.

Sincerely yours,

Dr. De La Flor José

Corresponding author's address